# First Results of Peer Training for Medical Staff—Psychosocial Support through Peer Support in Health Care

**DOI:** 10.3390/ijerph192416897

**Published:** 2022-12-16

**Authors:** Dominik Hinzmann, Marion Koll-Krüsmann, Andrea Forster, Andreas Schießl, Andreas Igl, Susanne Katharina Heininger

**Affiliations:** 1Department of Anesthesiology and Intensive Care, Klinikum Rechts der Isar of the TU Munich (TUM), 81675 Munich, Germany; 2Association for Psychosocial Competence and Support in Acute Care-PSU-Akut, 81373 Munich, Germany

**Keywords:** employee health, peer support, peer training, peer mental health, prevention in the working environment

## Abstract

Background: In view of the increasing strain on health workers, psychosocial support measures are becoming more important. The core of a sustainable concept is the establishment of peer support teams. Two aspects are central: first, target group-specific training content, and second, suitable staff members who are trained as peers. The goal of the study was to obtain a first look at what content can be taught in peer training for medical staff, how the training is evaluated by the target group, and which people are interested in training from peers. Methods: During the period 2017–2022, Peer Training for medical staff was developed by a non-profit institution in Germany with state funding and the support of a medical professional association and evaluated during the project. Participants (N = 190) in the Peer Training course were interviewed in advance about their experiences and stresses at work using an anonymous questionnaire. After completing the training modules, the participants filled out an evaluation form. Results: The participants of the Peer Training were predominantly female (70.5%) and middle-aged (between 31 and 50 years old). Most (80.3%) experienced stressful events themselves, mostly without any preparation (93.5%) or follow-up (86.8%) by the employer. The participants estimate their workload in the medium range. The proportion of stressed individuals among the participants was below that of various comparison groups as available reference values. The training module itself was evaluated very positively. Conclusions: The content and framework parameters of the training were rated very well. There is a high degree of fit with the requirements in the health sector. The participants in the Peer Training seem to represent a good cross-section of the target group medical staff, also regarding their own experiences, seem to have a good psychological constitution and are therefore very suitable to work as peers after the training.

## 1. Introduction

Medicine, especially acute medicine as a high-risk area, always required the highest level of professionalism from medical staff even before, but especially during, the COVID-19 pandemic. High demands and complex clinical pictures repeatedly present staff with difficult challenges. These often lead to acute stress and can result in chronic overload. In addition, particularly critical or potentially traumatizing events can occur at any time [1,2]. These can cause acute psychological impairments, intense emotions, and unsettling thoughts even in routine employees. These aspects have led to the consideration and research of so-called “second victims” among medical staff. A large body of literature is available on this subject (e.g., [3,4,5]).

It is therefore not surprising that an increased vulnerability to psychological stress can be found in the medical profession [6,7]. The suicide rate among physicians is also three to four times higher than in the general population [8,9,10], and in the field of anesthesia it is up to six times higher. Also in the field of anesthesia, it has been shown that numerous serious events are experienced in the course of professional activity, which has the potential to result in acute or chronic stress reactions [11]. In addition, in different samples, increased substance abuse has been shown in 10–15% of respondents [12]. Likewise, an above-average level of sick leave in nursing gives pause for thought [13], which can be assumed to be related to high burnout figures: A systematic review [14] found prevalences for burnout of staff in intensive care units of up to 47%, depending on the study.

Strengthening the resilience of employees in the run-up to stress (primary prevention), providing psychosocial support after special or extreme stress (secondary prevention), and ensuring functioning support functions and employee-friendly management structures (prevention on the organizational level regarding working conditions) are the central and decisive access routes to maintaining and promoting employee health and performance [15]. Therefore, it is also of particular importance that the future, collegial companions are able to move in a stable and resilient manner in the health medical setting described. Given this background, a training course for medical staff was developed to train colleagues to become peers for psychosocial support. The aim of the present study was to obtain a first insight into what content can be taught in peer training for medical staff, how the training “becoming a peer for psychosocial support” is evaluated by the target group, and which persons are interested in becoming a peer.

The following sections describe the existing concepts that were used for development, the specifics that need to be considered for healthcare staff, and how the identified content can be translated into a training format.

### 1.1. For the Prevention of Job-Related Overloads and Trauma-Related Disorders

Other occupational groups can also be seen as high-risk groups regarding the development of job-related mental illnesses. More than 40 years ago, concepts were developed in the USA to support firefighters and rescue workers (see also [16]). The core element here was aftercare by teams in which so-called peers (collegial supporters) worked under the leadership of psychosocial specialists. 

In German-speaking countries, the term psychosocial emergency care for emergency forces (keyword PSNV-E = German abbreviation, therefore) was established through the work of Beerlage et al. (2006) [17] for support in non-police emergency response, i.e., in the preclinical area. Over the years, it has become clear that focusing solely on aftercare is not very effective; primary prevention is at least as important. For example, as part of a nationwide research project, a primary prevention training course for fire brigades was developed and evaluated, taught by peers as part of the basic training [18].

In other occupational groups, such as bank employees, it has also been recognized that there is a high need for support after extreme events, such as a bank robbery. In the field of German savings banks, for example, collegial first-aid counselors after bank robberies have been trained for about 20 years. Within the framework of this concept, it became apparent that the inclusion of managers is of central importance in banks and that target group-specific training concepts are necessary [19]. Of course, different professional groups as well as different entities are not unrestrictedly comparable. Within the topic presented here, there are basically focuses on Second Victim, Moral Injury, Moral Distress, Moral Stress, Psychological Stress, PTSD, Burnout, Depression, and many other entities. These cannot be readily treated as a single entity and may have different strategies to deal with. Therefore, training programs that educate peers should make some distinction and can be seen as “first line” help for health care providers before referral to a specialist. Nevertheless, basic approaches to dealing with serious events can be drawn from other contexts for consideration and understanding.

Surprisingly, in Europe, with the exception of Switzerland, there are hardly any systematically described procedures to support medical or clinical staff, although the necessity is evident and also demanded [20]. Such projects are now established and studied in the Anglo-American region. For example, a peer support system was established at the Brigham and Women’s Hospital in Boston (USA) in 2004 after an anesthesiological incident [21].

### 1.2. Psychosocial Support in the Health Sector

The need for peer support in health care has been recognized in Germany primarily by two associations (SbE-Bundesvereinigung e.V. and PSU-Akut e.V.) in the last decade and the implementation of peer support in clinics has been strived for. The PSU-Akut e.V. association, founded in 2013, pursued a comprehensive, sustainable, and target group-oriented concept from the very beginning, in which particular attention was paid to the training of peers and the process of implementing peer support in order to do justice to the need for primary prevention and the prevention on the organizational level regarding working conditions [22].

Moreover, the conditions regarding serious incidents in health care differ in numerous factors from other activity-related serious incidents, such as violence in facilities with public traffic. These differences must be reflected in the training in order to achieve a fit with the target group.

### 1.3. Collegial Support within the Framework of Psychosocial Support

What are the tasks of peer supporters, and why the term peer support? “Peer group” is an expression that is also known in the German-speaking world. “He belongs to my peer group, I can tell him”. I trust someone who belongs to my peer group, who is close to me, has similar views, thoughts, and feelings as I do. 

In the context of job-related stress, the peer is also a peer among equals, he has “stable smell” and knows the issues from his own experience. Psychological support, as was also the experience in the COVID-19 pandemic, is often not called upon, even if it is offered [23]. This may be due to fear of pathologizing or stigmatization. In the area of psychological interventions for pre-existing symptoms, a meta-analysis found that interventions were most effective for psychological symptoms such as depression and post-traumatic symptoms, but were less helpful specifically in reducing general feelings of stress [24].

Peers are co-workers, they are on the same level, but have additionally qualified for peer support in their own organization. They are available as resources and contact persons at a “low-threshold” level in the case of stressful events and function both as multipliers of appropriate action knowledge and as an interface for the referral of additional support services. In Germany, for example, occupational accident insurance funds finance psychotherapeutic treatment if required. This is because potentially traumatizing events, like other injuries, are to be regarded as occupational accidents and are thus insured by the organization’s accident insurer in the same way as physical rehabilitation measures. 

In many cases, however, psychotherapeutic treatment options are taken up too late in the context of activity-related traumatization in the health sector. Experience shows that the support provided by the accident insurance funds does not reach those affected, or reaches them too late, with corresponding consequential costs for the organization and the accident insurance fund. For this reason, peers play an important role as guides, in addition to providing concrete support when needed. Ideally, there is contact with a peer before a traumatic experience because trust has already been built up, including the trust to be able to fall back on the peer’s knowledge about further help if necessary. 

For this to happen, peers must be known in the organization. It must be clearly visible that leaders support the peer support system. If an extreme, potentially traumatizing event occurs, peer support should take place promptly, be sustainable, always be offered repeatedly, reach all those affected if possible, and build on primary prevention events that are to be held regularly. It has been proven that collegial support as a low-threshold offer with barrier-free access reaches the target group well [25]. Managers should also be involved. Training them specifically for this purpose has proven to be effective.

### 1.4. Structure and Contents of the Peer Training at PSU-Akut e.V.

The Peer Training evaluated here comprises a multi-level training concept for medical and therapeutic staff in the health sector. In total, the training comprises 5 days, one day contains nine teaching units (UE) of 45 min each. In addition to the “classic” contents of Peer Training, the curriculum includes specific training contents adapted to the target group. Examples include the following topics: dealing with feelings of guilt, legal issues, and difficulties in communicating with patients and relatives. The curriculum is designed for the target group of doctors, nurses, medical assistants, and therapists.

The basic building block is two days, referred to as Module I in the following. Module II, which also lasts two days, is mainly relevant for people who will work as peers not only with individuals but also with groups and teams. Module III, also called the multiplier module, is primarily for primary prevention; here the peers who have already completed Module I and Module II receive knowledge about the internal communication of peer support in their institutions. Peers who become active as part of a peer support structure often attend modules I and III in combination. 

Corresponding training structures have been developed by the association PSU-Akut e.V. and are offered as an open offer as well as in-house training on demand, for those who want to build an entire peer team right away as an entire organization (clinic, care facility, etc.). Other institutions can also book the peer training as a whole or single modules for training

The complete training thus takes place over a total of three modules. In each module, practical exercises and behavioral training take place in addition to theory. 

In addition to the training, PSU-Akut e.V. offers regular supervision units for already trained peers. The peers can contact a telephone hotline, the PSU-HELPLINE, with questions about interventions in their facilities, in order to receive prompt further support from PSU experts and, if necessary, also request personnel support for group interventions on site. In addition, the PSU-HELPLINE can also be used as a fallback level for possible own stresses from the peer activity. The PSU-HELPLINE is staffed 7 days a week with peers as well as its own fallback level on their part. Callers have the option to make the call anonymously if required.

#### Contents of the Teaching Modules

*Intervention-Module I (18 units of 45 min).* The aim of this module is to provide participants with basic skills in dealing with stress in the context of serious events and phases of stress as well as in peer support. The focus here is on acute intervention services with one or two affected persons. Focal points are, for example, psychosocial support and starting points for prevention, coping and stress regulation, and conducting conversations with those affected.

*Intervention-Module II (18 units of 45 min).* The aim of this module is to provide participants with basic competences relevant to practice in leading discussions and peer support. The focus is on crisis intervention services for affected teams or groups. The prerequisite for participation in Intervention Module II is successful participation in Module I. The focus of this module is, for example, further help with trauma sequelae, working with teams/groups, and providing brief psychosocial support directly and a few days after an event.

*Module III-Multiplier Module (9 units of 45 min).* The multiplier module aims to support trained peers and psychosocial professionals in conducting information events and short training sessions and to provide suitable working and information materials for this purpose. Together with the participants, cases of application are discussed, and concrete implementation possibilities are worked out and practically tested. Participation in the multiplier module requires at least participation in Module I or another training in psychosocial support.

The focal points of this module are, for example: planning information events and short trainings, working out concrete implementation possibilities, and working with a multiplier toolbox. For further insight into the content and didactic realization of the training, please contact info@psu-akut.de.

## 2. Materials and Methods

### 2.1. Procedure and Research Questions

For the present study, the results of Module I of the Peer Training were evaluated. The present study was conducted in a cross-sectional design. The present study emerged from accompanying research on peer training. The participants of the peer training represent the sample. The data collection took place in the period from November 2017 to the beginning of May 2022. Only with the onset of the COVID-19 pandemic was the GHQ-12 used as a supplement in the surveys, so the sample size is different. Questionnaires were offered before (focus: items on the person) and after the training (focus: items on the training) to split the working time.

The following questions were guiding:Do the developed training contents fit the stress conditions and coping requirements in the health care system?Which group of people decides to be trained as a peer?Do the peers bring the desirable psychological robustness?How do the participants assess their current workload?

The questionnaires contained items and scales on the following topics:Information on the participants (including socio-demographic data);Previous experiences with emotionally stressful and serious events in the work context as well as the individual way of dealing with them;Psychological hazards and stress at work;Perceived workload;The psychological stress of the participants;The individual’s “general mental health”;The assessment of the individual ability to recover from stress despite adversity.

Written participant feedback was obtained following Module I of the Peer Training.

### 2.2. Research Instruments and Evaluation Methodology

All analyses were purely descriptive and were conducted using IBM SPSS Statistics 28. The collection of socio-demographic data, as well as previous experiences with emotionally stressful and serious events and how they were dealt with, was carried out by means of a questionnaire specially designed for the survey. Established and validated measures were used to assess mental health and workload. These are described below. 

#### 2.2.1. General Health Questionnaire

The General Health Questionnaire (GHQ-12) [26] is a reliable instrument that can be used to identify psychological stress. It consists of 12 items that ask about the general state of health in the past weeks. For example, it asks: “Did you sleep less during the last few weeks because of worries?” or “Have you been able to carry out your daily responsibilities with pleasure in the last few weeks?”. 

The questions could be answered on a 4-point Likert-Scale (coding 1 to 4). Positively worded items increase the GHQ score if they are answered with “worse or much worse than usual”. Negatively worded items increase the GHQ score if they are answered with “more or much more than usual”.

For the evaluation of the GHQ-12, all responses coded 1 and 2 were first recoded to a 0, and all responses coded 3 or 4 were recoded to a 1. After eliminating cases in which there were many missing values, a sum was formed from the 12 items with the new coding. Subsequently, the participants could be divided into groups by forming the corresponding variables: 

Group 1 (scores 0 to 4) means that there is no or low psychological stress and group 2 (scores 5 to 7) contains participants with medium psychological stress. Participants with high psychological stress are assigned to group 3 (scores 8–12).

#### 2.2.2. Workload

The scale “Workload in Nursing” [27] depicts the subjectively perceived workload of nursing staff. For the use of the scale in the questionnaire for Peer Training, the term “doctors and staff” was adapted to “colleagues” and thus found application for all participants from the different professional groups.

The scale range is from 1 to 5 (Likert scaled), with higher values corresponding to a higher overall workload, greater coordination and information problems, and greater psychophysical overload.

Due to the calculation of Cronbach’s alphas and higher reliability, item 9 was not included in the calculation of the variables in the workload total as well as in the coordination and information problems subscale and item 7 in the psychophysical overload subscale. The workload scale is reliable. The Cronbach’s alpha value for the items used to measure workload is α = 0.79 (Subscale Coordination and information problems: α = 0.74, Subscale Psychophysical overload: α = 0.73).

Following the formation of variables, the mean values, standard deviations, min., and max. were calculated for both subscales and for the entire scale.

#### 2.2.3. Aspects of Mental Risk Assessment

In order to get an impression of the mental stress at the workplace, based on the assessment criterion “mental stress” of the risk assessments for companies required by law in Germany according to § 5 of the Occupational Health and Safety Act (Act on the Implementation of Occupational Health and Safety Measures to Improve the Safety and Health Protection of Employees at Work), a short scale with 5 items was developed, which asks for a quantitative assessment of the following four dimensions mentioned in the mental risk assessment on a 10-point Likert-Scale: “work task/content”, “work organization”, “social/company conditions”, “workplace/environmental conditions”. Instead of the fifth dimension “forms of work” (telework, fixed-term contracts, etc.), the scale was extended by an item on “stressful events”. As an instruction, the participants were asked the following question: “Please now consider which of the following areas have been particularly stressful for you recently and tick the appropriate box for the individual aspects”. Higher point values correspond to higher stress.

#### 2.2.4. Participant Feedback

The evaluation offers the opportunity to assess the framework conditions of the seminar and the extent to which the expectations of the seminar were met. Furthermore, it is asked which further psychosocial support options the participants would like to have. The participants could rate these aspects on a scale from 1 to 5 (1 = very poor to 5 = very good).

### 2.3. Research Ethics

In order to protect their data and to take into account the aspects of voluntariness and informedness, training participants were actively asked by the instructors for permission to store data and informed that they would not suffer any disadvantages if they did not agree to the data collection. A corresponding notice was also given in writing and handed out to the participants together with the questionnaires. In accordance with the regulations of the German Data Protection Ordinance, the option of entering a pseudonymization code was given, also in order to enable the deletion of one’s own data record if necessary. 

For this survey and evaluation, an ethics vote (683/20 S-SR) of the ethics committee of the Klinikum rechts der Isar of the Technical University of Munich, Munich, Germany, is available.

## 3. Results

The results of the surveys relevant to the first question are presented below. 

A total of N = 351 participants attended the described intervention modules I and II as well as the multiplier module. N = 190 persons took part in Module I. Of these, N = 179 data sets were available. The following results refer to this sample.

### 3.1. Socio-Demographic Data of the Participants

Of the participants, 70.5% were female and 29.5% male. The age structure was heterogeneous, with most participants (56.5%) being between 31 and 50 years old. A total of 29.8% were in the 51 to 70 age group. The younger participants are proportionally the smallest group: 13.7% were 30 years and younger.

Figure 1 shows the distribution among the different professional groups.

Due to the heterogeneity in terms of age, the distribution of years of work is also dispersed. (see Figure 2): The largest share, 22.1%, was made up of participants who had been working for 26 to 35 years.

Most participants worked in intensive care units in hospitals (32.5%), followed by employees who worked in areas that can be grouped under “others” (18.7%). These include, for example, occupational medicine, the staff council, or pastoral care. A total of 16.3% of the participants worked in emergency rooms in hospitals and 13.8% in normal wards in hospitals. In addition, staff from the areas of anesthesia (7.3%) and operating rooms (5.7%) as well as from rescue services (pre-clinic) (2.4%), medical practice (1.6%), and delivery rooms (1.6%) took part. (Figure 3).

Of these, 70.2% worked full-time and 29.8% part-time. A total of 39.7% were working as managers at the time of the survey. A significantly larger proportion did not hold a management position (60.3%).

Further, 72.0% stated that they had acquired an additional qualification. These include, for example, the additional title of “emergency doctor” or further training in specialist nursing, e.g., in the areas of anesthesia/intensive care or emergency nursing.

A total of 59.6% worked in maximum care level hospitals, 19.8% in intermediate care level hospitals, and 20.6% in basic care level hospitals (see Figure 4).

### 3.2. Descriptive Data of the Items and Scales

#### 3.2.1. Mental Health (Results of the General Health Questionnaire, GHQ-12)

The participants of the Peer Trainings from whom complete answers for the GHQ-12 scale were available (n = 101) were analyzed according to the evaluation criteria. The data show the following distribution (see Figure 5): 77.2% of the participants were assigned to group 1 (no or low psychological stress), 9.9% were assigned to group 2 (medium psychological stress) and 12.9% were assigned to group 3 (high psychological stress).

#### 3.2.2. Workloads at the Workplace

Overall, the participants (n = 129) rated their total workload with a mean of 2.96 (SD 0.47) (range from 1 to 5 on a Likert-Scale). The subscale “Coordination and Information Problems” had a mean of 2.90 (SD 0.56). The subscale “Psychophysical Overload” was rated with a mean of 3.16 (SD 0.64) (see Figure 6).

#### 3.2.3. Aspects of Mental Risk Assessment

The mentioned aspects of the mental risk assessment were assessed by the participants as follows in Figure 7 (work task n = 114, work organization n = 112, social relations n = 113, work environment n = 113, and stressful events n = 112). Accordingly, the aspects of work organization and work environment seem to be perceived as particularly stressful. The participants seem to be able to cope well with the aspects of social relationships and burdening events.

#### 3.2.4. Individual Experiences with Stress and Support Services

A total of 80.3% of the participants stated that they had experienced dramatic and emotionally very stressful events at work in the last 2 years. Further, 60.3% report that there is a point of contact for this in their clinics; 93.5% reported that they were not prepared by their employer for possible extreme events and resulting consequences; and 86.8% stated that after a stressful event, the employer did not help with emotional-psychological processing. Of those who had experienced support from the employer, 15.6% indicated a collegial discussion and 12.3% a discussion with the supervisor as a support format, which were the largest categories.

#### 3.2.5. Evaluation of the Peer Training

The Peer Training was rated on a Likert scale from 1 to 5 (1 = very poor to 5 = very good) as very good by the participants (n = 130) for both the content taught and the framework conditions (see Figure 8).

## 4. Discussion

In this study, a total of N = 190 participants in Module I of a Peer Training program for medical and therapeutic staff in the health sector were interviewed about their experiences and stress at work. From the beginning of the COVID-19 pandemic, participants also completed a questionnaire on psychological stress. An evaluation of the training module was also collected. In summary, the questions are discussed below in relation to the results found for module 1 for N = 179 data sets.


*Regarding question 1: Do the developed training contents fit the stress conditions and coping requirements in the health care system?*


The information provided was rated on average at 4.57 on a five-point scale. Thus, it can be assumed that the training content provided fits the stress conditions and coping requirements. 

The following aspects should be mentioned as limiting:

Not all participants in the PSU training could be motivated to complete the questionnaires with full responsibility. It would also be desirable to analyze the training contents in more detail. One can also wonder whether social desirability could have influenced the results, even though the survey was conducted pseudonymously.


*Regarding question 2: Which group of people decides to be trained as a peer?*


It could be shown that the group of people who decide to be trained as a peer approximately represents the entirety of health workers in terms of gender ratio. The participants in Peer Training are predominantly female (70.5%), which roughly reflects the actual distribution. In 2021, for example, around 83% of nursing staff in Germany were women [28]. However, the proportion of female doctors working in hospitals was also only 47% in 2020 [29], but has been growing continuously for years.

A peer is supposed to be an equal among equals.

The age structure is heterogeneous, although the age of most participants (56.5%) is between 31 and 50 years. Linked to this, the majority of participants have more than 10 years of working experience (71.4%) and more than a third (36.1%) have more than 20 years. The participants thus have a lot of experience, which can also be seen as a favorable parameter, because the safety and expertise of the peers have a positive effect on the outcome of the interventions [18].

From a pragmatic point of view alone, such as accessibility and time budget, the high number of peers working full-time (70.2%) can also be considered favorable. 

Most participants work in intensive care units in hospitals (32.5%) or in emergency rooms (16.3%). A large proportion does not hold a management position (60.3%). Most (72.0%) of the participants have an additional qualification. The majority (59.9%) of the participants also work in maximum-care level hospitals. These variables can also be seen as favorable for the work as a peer.


*Regarding question 3: Do the peers bring the desirable psychological robustness?*


The discussion of the results is related to figures on the burden in the population measured with the GHQ in the period mentioned, whereby in the absence of German GHQ data in the first year of the pandemic, a British survey is used. The working group around Pierce (2020) [30] examined the general population in Great Britain in April 2020. They found that more than a quarter (27.3%) of the population reported a GHQ-12 score that indicated a clinically significant level (more than 4) of psychological distress. In the present sample, only 22.8% of the people reported a clinically significant level (more than 4) of psychological distress. 

Although it can be assumed that the peers were under enormous stress due to their work in the health sector, they show significantly lower values in relation to the GHQ than in the British survey of the general population at the time of the pandemic, although they are certainly higher than those measured there in the years before the pandemic (around 19%). However, the higher proportion of women in the peer sample must also be taken into account, because women generally achieve higher values, which can also be seen in the British sample. 

The data found can also be compared with other studies that recorded mental stress in the health sector during the pandemic. Greenberg et al. (2021) were able to show that 40% of participants from the intensive care sector were above the clinical cut-off for post-traumatic stress [31]. A high correlation between psychological stress in general, measured with the GQH, and trauma-relevant symptoms was shown in a study of volunteer fire brigade personnel [18].

It can also be concluded from the comparison with stress values from the health sector that the group of people who are trained as peers have good psychological robustness and resilience, which is an essential prerequisite for their work as peer supporters in the health sector. 

This is also supported by the results of question four. 


*Regarding question 4: How do the peers assess their current workload?*


Most of the participants in the Peer Training (80.3%) have their own previous experience with stressful events. They report that although there are often contact points for this in the hospitals (60.3%), they report, to the contrary, that the employer does not provide support in dealing appropriately with stressful events and overloads: not in the preparation (93.5%) and not in the follow-up (86.8%).

Despite the individual previous experiences, the participants seem to be able to cope well with challenges inherent to their job. On average, the participants state a medium perceived workload and seem to be able to deal comparatively well with the aspects of “social relationships” and “burden events” regarding the assessment of risks to mental health. The fact that especially the aspects “work organization” and “work environment” are perceived as stressful speaks for the high relevance of peer support accompanying measures of relationship prevention, i.e., concrete measures to improve the respective work situation. This assumption is in line with the findings of Blake (2020) [32], which describe that the establishment of so-called wellbeing centers (retreat and recreation rooms) was not only welcomed and used by the workforce but also the low-threshold peer support offered there.

### Limitations

Limitations to note are that the sample studied is not yet of a truly robust size and might show a selection bias (e.g., motivation to be trained, tuition fees for attendance, social desirability) or a gender bias. In addition, only initial descriptive analyses have been conducted so far. In this first analysis, we offered subjective ratings to the participants which can only be used to answer the question if the participants “feel” more informed, educated, and skilled instead of testing their competencies. Future research is planned to focus, e.g., on measurable objectives such as learning outcomes. For selected aspects, more in-depth analyses should provide an insight into the profiles of people who want to be trained as peers in order to strengthen the current findings that interested persons have a suitable personality profile/stress profile/coping profile (“which group of people decides to be trained” combined with the lower-than-average GHQ scores of participants may suggest that those who chose to participate had the psychological resources to take part”). 

Overarching developments with an influence on the health care system can be mentioned as limitations. In the period between 2017 and 2020, there may have been different influencing factors that affected the willingness to participate in peer education, the experience of stress, increasing staff shortages, or other factors.

For the question of which people a clinic/company should select to be trained as a peer, there are currently only empirical values (please contact www.psu-akut.de). Further studies are planned.

## 5. Conclusions

The literature available to date clearly shows how high the stresses and their negative consequences are among medical staff. Available evidence also shows, as do the results of the present study, that employers hardly offer effective support measures so far. Against this background, it is of great importance to identify and implement low-threshold support options that can be well accepted by medical staff without having to fear negative consequences for reputation or professional development. One possibility is to train peers in psychosocial support in clinics or health care facilities and have them work in and with the teams. 

The present work contributes to increasing the knowledge about the group of people who are trained as peers. This knowledge is important in order to draw conclusions about the quality of the peer support structures in the institutions. Furthermore, parameters can be derived from this knowledge if institutions want to select staff members specifically to have them trained as peers. The results show that the training participants appear to be predominantly psychologically robust and resilient, and despite a high workload, as was the case in the times of the pandemic, as well as witnessing stressful events, they are on average less psychologically stressed, this in comparison with the general population as well as other people working in the health sector. Thus, it can be assumed that staff members who are trained as peers have implicit coping knowledge and do not belong to a particularly stressed group that might decide to undergo such training in order to better cope with their own issues. 

The present results can help to obtain a first insight into which contents can be taught in peer training for medical staff, how the training “becoming a peer for psychosocial support” is evaluated by the target group and which persons are interested in peer training. The contents developed by PSU-Akut e.V. for “Peer Training for medical staff” (in addition to the described training contents within the framework of collegial support) are judged to be suitable by the peers. Particularly in the healthcare sector, the special and complex psychological hazards and their coping conditions must be taken into account in the context of Peer Training.

For complementary research approaches, the measures carried out by the trained peers in their institutions could be evaluated in the next step. This could provide further insights into the fit and quality of PSU training.

## Figures and Tables

**Figure 1 ijerph-19-16897-f001:**
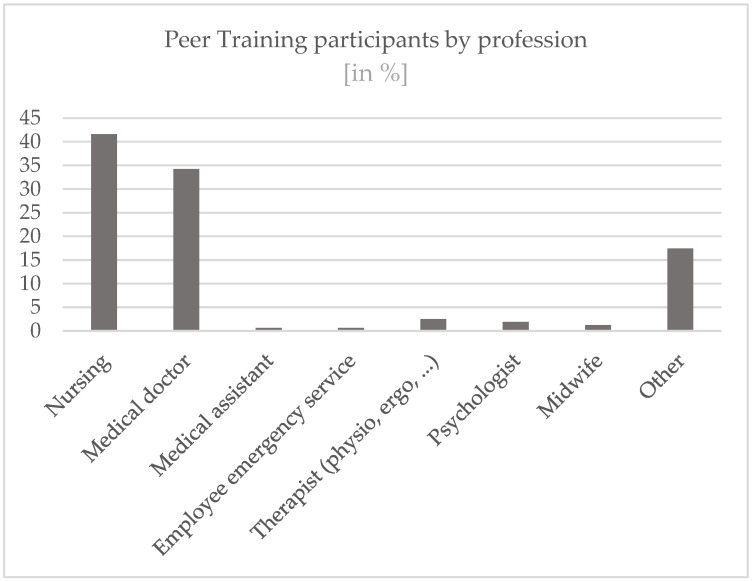
Profession details.

**Figure 2 ijerph-19-16897-f002:**
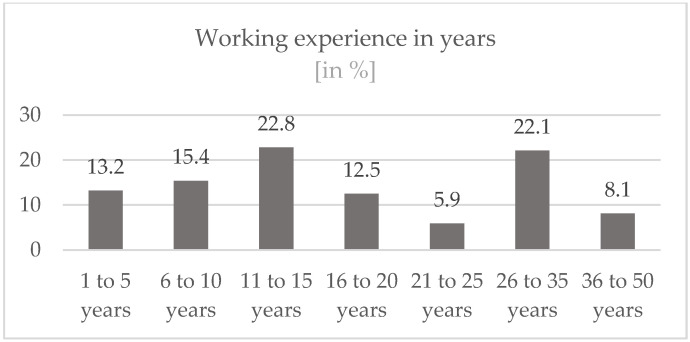
Working experience in years.

**Figure 3 ijerph-19-16897-f003:**
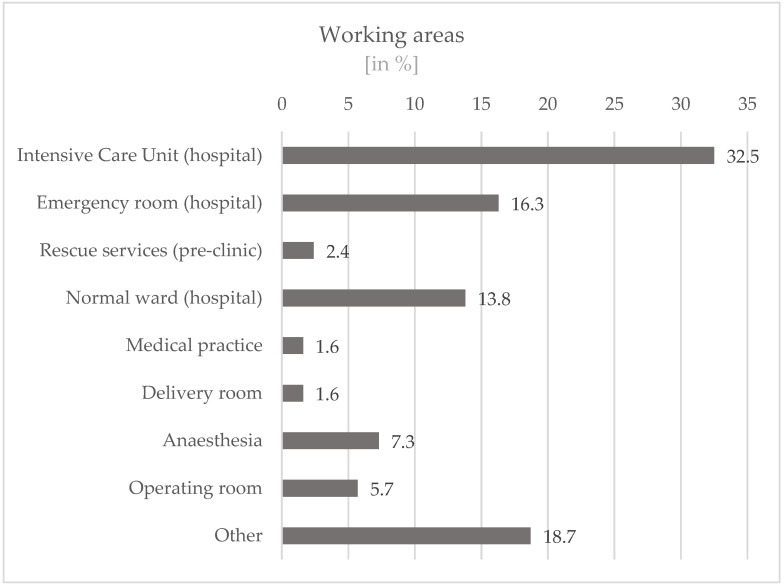
Working areas of the participants.

**Figure 4 ijerph-19-16897-f004:**
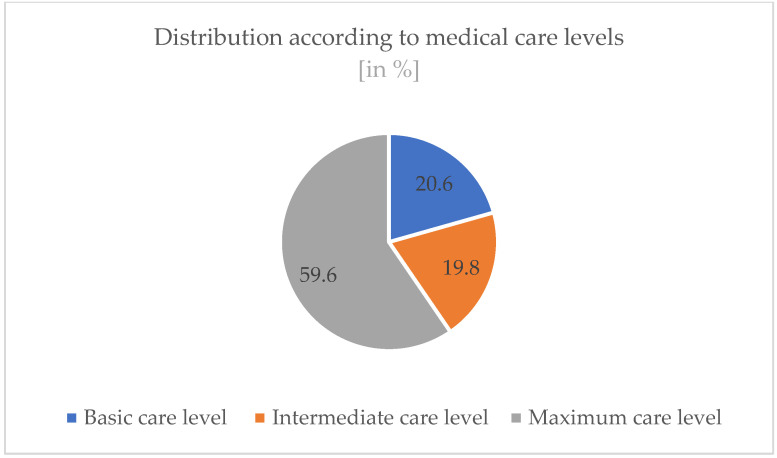
Medical care levels of the participants’ organizations.

**Figure 5 ijerph-19-16897-f005:**
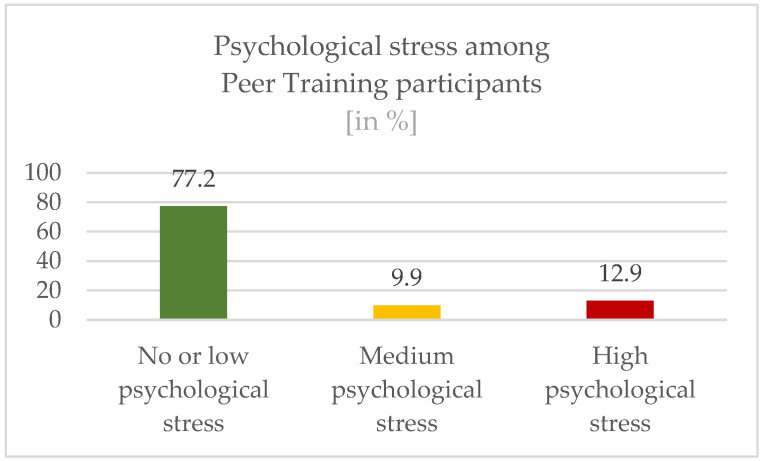
Results of the General Health Questionnaire among participants.

**Figure 6 ijerph-19-16897-f006:**
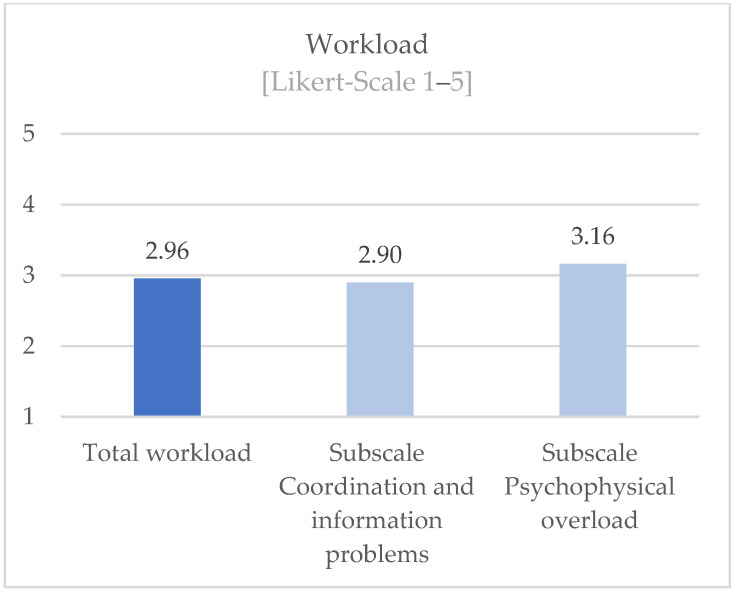
Perceived workload among participants.

**Figure 7 ijerph-19-16897-f007:**
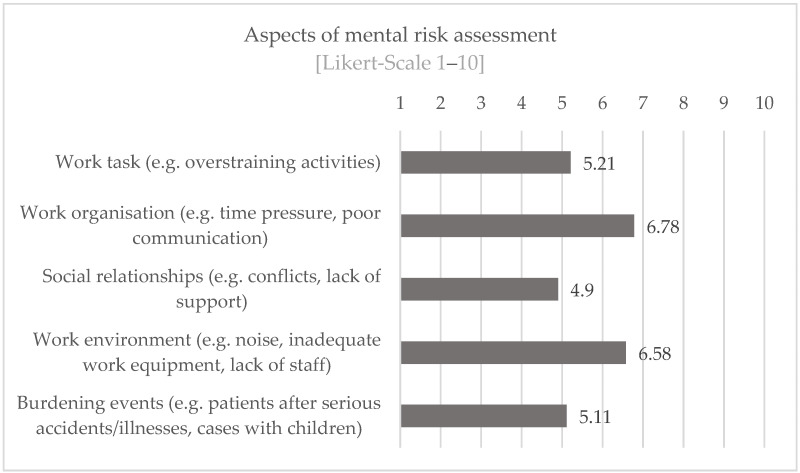
Assessment of workload through aspects of mental risk assessment.

**Figure 8 ijerph-19-16897-f008:**
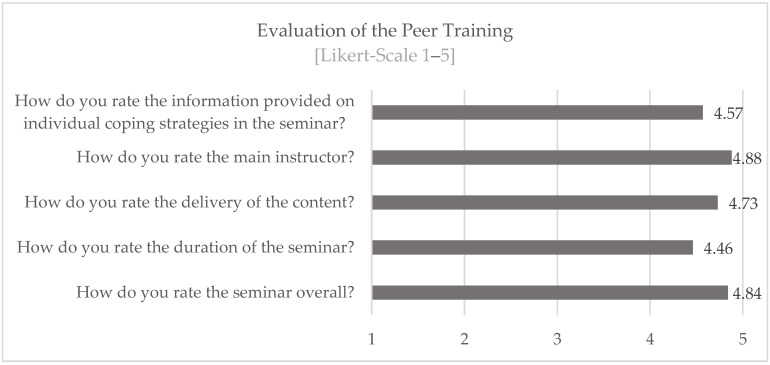
Evaluation of the Peer Training.

## Data Availability

Not applicable.

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
