# Peer review of "First Results of Peer Training for Medical Staff—Psychosocial Support through Peer Support in Health Care"

_ijerph, 2022, doi:10.3390/ijerph192416897_

Round 1
Reviewer 1 Report
This article presents survey data from various healthcare workers (ranging from a couple medical assistants, psychologists and midwives to more representation from nurses and medical doctors, across specialties) on various factors relating to their well-being. Participants completed a peer support training course and were surveyed following course completion. The training modules sounded novel and were perceived as valuable by the participants.
The introduction was well-written and compelling, regarding the importance of the need for new methods of supporting healthcare worker well-being. However, I was confused about the focus of the article, due to both the abstract and article focusing more on the reported well-being related factors and demographics than on the effects of the training (which was what I initially thought the article would be about). Is the goal primarily to describe current state of well-being for those who elect to be peer supporters, or more about the value of such a training?
In order to make it more useful to readers, it would be helpful for the authors to provide logistical details about how they implemented the program. For example, what financial resources were required for the staffing Help line? Are the modules available for those at other healthcare institutions who would like to build their own programs (or pay for the already developed curricula)?
The lack of a limitations section is striking, and should be addressed. In the limitations section, it would be important to note potential reasons for the patterns observed, and the inability to determine causation from the cross-sectional design. For example, the question of "which group of people decides to be trained" combined with the lower-than-average GHQ scores of participants suggests that those who chose to participate had the psychological resources to take part. It would be helpful if the authors could suggest how other institutions would want to recruit peer support based on the results of these data.
I thought the results presented on lines 341-345, regarding the high rates of participants experiencing very stressful events at work and not feeling prepared for these by the employer, deserved more attention in the conclusion, along with a call to action (or at the very least, suggestions for healthcare leadership to begin to address this). At the end of the article, I was still left confused about the purpose of this manuscript: is it to discuss the value of peer training or merely document current state? If the authors took more of a stand on how the reader should make use of the data presented, and perhaps included a "future directions" section, that would be helpful in clarifying what readers should be taking away.
Author Response
Article: Peer Training for Medical Staff - Psychosocial Support through Peer Support in Health Care
Dear Reviewer,
Thank you very much for your review and your valuable and justified comments and suggestions.
Please find below our responses and adjustments to your comments and queries:
Comments and Suggestions for Authors
This article presents survey data from various healthcare workers (ranging from a couple medical assistants, psychologists and midwives to more representation from nurses and medical doctors, across specialties) on various factors relating to their well-being. Participants completed a peer support training course and were surveyed following course completion. The training modules sounded novel and were perceived as valuable by the participants.
The introduction was well-written and compelling, regarding the importance of the need for new methods of supporting healthcare worker well-being. However, I was confused about the focus of the article, due to both the abstract and article focusing more on the reported well-being related factors and demographics than on the effects of the training (which was what I initially thought the article would be about). Is the goal primarily to describe current state of well-being for those who elect to be peer supporters, or more about the value of such a training?
Reply to the author: Thank you very much for your question. For better understanding we have added the following lines to Abstract (line 15ff): "The aim of the study was to get a first look at what content can be taught in a peer training for medical staff, how a training "to become a peer for psychosocial support" is evaluated by the target group and which people are interested in becoming a peer."
and Introduction (line 62ff): “Given this background, a training course for medical staff was developed to train colleagues to become peers for psychosocial support. The aim of the present study was to get a first insight into what content can be taught in a peer training for medical staff, how a training "becoming a peer for psychosocial support" is evaluated by the target group and which persons are interested in becoming a peer. The following sections describe the existing concepts that were used for development, the specifics that need to be considered for healthcare staff, and how the identified content can be translated into a training format.”
In order to make it more useful to readers, it would be helpful for the authors to provide logistical details about how they implemented the program. For example, what financial resources were required for the staffing Help line? Are the modules available for those at other healthcare institutions who would like to build their own programs (or pay for the already developed curricula)?
Reply to the author: Thank you for your questions to clarify the context. We added the following lines:
- Line 171f: “The PSU-HELPLINE is staffed 7 days a week with peers as well as its own fallback level on their part. Callers have the option to make the call anonymously if required.”
- Line 160ff: “Corresponding training structures have been developed by the association PSU-Akut e.V. and are offered as an open offer as well as in-house training on demand, for those who want to build an entire peer team right away as an entire organization (clinic, care facility, etc.). Other institutions can also book the peer training as a whole or single modules for training”
The lack of a limitations section is striking, and should be addressed. In the limitations section, it would be important to note potential reasons for the patterns observed, and the inability to determine causation from the cross-sectional design. For example, the question of "which group of people decides to be trained" combined with the lower-than-average GHQ scores of participants suggests that those who chose to participate had the psychological resources to take part. It would be helpful if the authors could suggest how other institutions would want to recruit peer support based on the results of these data.
Reply to the author: Thank you for these comments and advice. We added a section “Limitations” in line 450f.
I thought the results presented on lines 341-345, regarding the high rates of participants experiencing very stressful events at work and not feeling prepared for these by the employer, deserved more attention in the conclusion, along with a call to action (or at the very least, suggestions for healthcare leadership to begin to address this).
Reply to the author: Thank you for this recommendation. We added the following lines (line 462ff) in conclusions. “The literature available to date clearly shows how high the stresses and their negative consequences are among medical staff. Available evidence also shows, as do the results of the present study, that employers hardly offer effective support measures so far. Against this background, it is of great importance to identify and implement low-threshold sup-port options that can be well accepted by medical staff without having to fear negative consequences for reputation or professional development. One possibility is to train peers psychosocial support in clinics or health care facilities and have them work in and with the teams.”
At the end of the article, I was still left confused about the purpose of this manuscript: is it to discuss the value of peer training or merely document current state? If the authors took more of a stand on how the reader should make use of the data presented, and perhaps included a "future directions" section, that would be helpful in clarifying what readers should be taking away.
Reply to the author: Thank you thank you very much for this remark. We added the following lines to close the loop (line 482): “The present results can help to get a first insight into which contents can be taught in a peer training for medical staff, how a training "becoming a peer for psychosocial sup-port" is evaluated by the target group and which persons are interested in a peer training.”
We hope you understand our adjustments, thank you again for your efforts and support, and remain available to answer any queries you may have!
Kind regards,
Dominik Hinzmann, [email protected],

Reviewer 2 Report
Dear editor and authors.
Thank you for the opportunity to review the manuscript by Hinzmann et al titled “Peer Training for Medical Staff – Psychosocial Support through Peer Support in Health care” submitted to the IJERPH: special issue on second victim.
To my eyes this is a very important contribution to strategic intervention on coping strategies concerning the Second Victim phenomonen (SVP) and other psychosocial factors affecting health care crews.
However, the presentation of the study should be improved at several points as it seems that only a minor proportion of the potential of the study results are presented in this manuscript:
In detail:
INTRODUCTION
Well written and giving a good overview. However, the term "second victim" is not mentioned (in a special issue about svp). That may be amended. Further the buddy study by Katja Schroeder from Denmark is a similar project that could be referred to in the introduction or discussion.
Further the recommendations of the AHA after events in cardiopulmonary resuscitation with recommended debriefings after ACLS and PALS cases also are substantial to detect and treat adverse reactions ("the wounded healers").
In Line 74 there is a reference to bank employees suffering to PTSD-like symptoms. However, there is a bundle of phenomena and diseases that all can be summarized under the term "psychosocial burden". There should be some clarification about the different entities that are presented together and addressed by the study program: Second victim, moral injury, moral distress, moral stress, psychological stress, PTSD, burn-out, depression (that has to be diagnosed by a psychiatrist) and many other entities belong to this group and cannot easily be treated as one and may have other coping strategies. Thus, training programs educating peers should give some distinction about this and should be presented more stronger as a "first psychological aid" to health care providers before transfer to a specialist.
Line 88: please explain the abbreviations
Line 135: in the manuscript there is a short explanation of the curriculum. As the manuscript is about a didactic format the readers do not get much information about the curriculum and its teaching formats itself. Consequently, a reproduction of the study would not be possible. Please consider distinct and structured information (e.g. using the Kerns Analysis or another curriculum developing tool) addressing the "Problem" (this is pointed out), "The Requirements" (What do we need and want?), "Objectives of the Curriculum" (What are the goals?), "Strategies of teaching" (how do we teach?), "Implementations issues" (Who is responsible? Who approves? Who supports? Who has to be informed?) , "Testing and Evaluations" (How and what do we test?) and "Dissimination" (How do we spread our work?).
Further, consider to present the curriculum blueprint, teaching and evaluation formats, time tables, needed materials, educators' qualifications and other information, e.g. in the appendices or supplementals to grant the possibility of repetition of the study and to reveal further insight for readers.
MATERIALS
Line 187: is this really a cross-sectional study with evaluation in a pre-post format (line 190)?
Please explain how the recruitment of the participants was done.
Line 209: You explained that you did a pre-post format (line 190) with use of a bundle of instruments. It is not clear why statistics is accomplished purely descriptive? With a pre-post examination e.g. by GHQ-12 and subjective workload and risks you would be able to do analytical tests on the development of the participants aside from subjective evaluations. If not possible, discuss this issue, as subjective evaluations are not sufficient and questionable to detect whether a training format is successful or not (depends on how do you define success: do people learn and get competent? Or do they think it was good?)
What were your hypotheses? What is tested? How was it tested? Please line this out for better understanding of your work.
RESULTS
As mentioned above the result section is minimalistic as only descriptive data is represented. If there was pre-post data, a significant amount of the results is lost in the present manuscript form. This epidemiological data is described satisfactorily (and is important), although the selection bias is not discussed.
Presentation of the data should be improved (why single columns and not boxplots?). With groups and the presented data on mean and SD effect sizes (Cohens D or even Cohens D-av for inter-, Cohens D-z for intra-individual calculations) are calculatable and relevant for educators, curriculum planers, quality managers and distributors.
Line 349
Ratings of the programs to be "good" or "Not good" (x-axis labeling is not explained) are highly (!) subjective and not robust for format or curriculum evaluation and therefore are not an indicator of effectiveness, efficacy, or efficiency of the program. Please discuss.
Further, there is no information whether learning objectives have been accomplished or not – if there were any outlined: what were the measurable (!) objectives and goals of the program? Were there tests on learning dimensions and competence such as factual knowledge, attitude, communication skills (see Bloom's taxonomy)? If not: discuss it for future and ongoing research as they are quality indicators of curriculum evaluation.
What Bias do you expect? To my eyes: selection bias (e.g. motivation to be trained, perhaps fees and social desirability.
Further, subjective stress from 2017 to the pandemic may have changed. Also, national regulations changed (eg. PPUGV in Germany on the one side, staff shortages of the other). Please discuss these variations. Has psychological burden changed in participants in 2017 to 2020 ?
Have there been subgroups effects, e.g. in the stress-groups or workload-groups? Age depend changes?
DISCUSSION
Throughout the discussion more external validity would be helpful for better stratification of generalization of some conclusions. However, as analytical data seems to be available, the discussion section should revisited when analytics were done. If not, this should be discussed in more detail.
Further, as the main hypotheses are not outlined, readers may be confused by this discussion as it is mainly epidemiological and focusing on descriptions but not on the learning format and the program that is the protagonist of the manuscript at the first view. Consequently, the heading should be adopted in case epidemiological findings alone are the aim of this study. Otherwise it may be distracting for some readers.
Line 406: Please discuss this gender bias.
Linie 414: This may be prone to a selection bias of the sample.
Why do you think that a low test results in a single GQH-12 is a valid and reliable parameter for resilience or psychological robustness? Are there studies on "robust" persons with the score? I think we should be careful, as psychiatrist that may be considered very robust can be second victims too and psychological stress may even effect robust people depending on the situation.
CONFLICT OF INTEREST
Please reevaluate any COI or bias arising from it as the format reported on is a "semi-commercial" product according to the web-page and the study group is the only provider of it.
As a conclusion: I think Hinzmann et al conducted a pioneering and promising approach to create a peer support program. Unfortunately, to my opinion this manuscript does not use its full potential.
The authors should consider presenting concise information about the curriculum, outline hypotheses, should do deeper analytics on available data, reevaluate graphics and calculations of effect sizes and should reevaluate the whole discussion and conclusion section after that.
Author Response
Article: Peer Training for Medical Staff - Psychosocial Support through Peer Support in Health Care
Dear Reviewer,
Thank you very much for your review and your valuable and justified comments and suggestions.
Please find below our responses and adjustments to your comments and queries:
Comments and Suggestions for Authors
Dear editor and authors.
Thank you for the opportunity to review the manuscript by Hinzmann et al titled “Peer Training for Medical Staff – Psychosocial Support through Peer Support in Health care” submitted to the IJERPH: special issue on second victim.
To my eyes this is a very important contribution to strategic intervention on coping strategies concerning the Second Victim phenomonen (SVP) and other psychosocial factors affecting health care crews.
However, the presentation of the study should be improved at several points as it seems that only a minor proportion of the potential of the study results are presented in this manuscript:
In detail:
INTRODUCTION
Well written and giving a good overview. However, the term "second victim" is not mentioned (in a special issue about svp). That may be amended. Further the buddy study by Katja Schroeder from Denmark is a similar project that could be referred to in the introduction or discussion. Further the recommendations of the AHA after events in cardiopulmonary resuscitation with recommended debriefings after ACLS and PALS cases also are substantial to detect and treat adverse reactions ("the wounded healers").
Reply to the author: Thank you thank you very much for this remark. We added the following lines (line 43ff): “These aspects have led to the consideration and research of so called “second victims” within medical staff. A large body of literature is available on this subject (e.g. [3–5]).
That includes the following references:
- Schrøder, K.; Bovil, T.; Jørgensen, J.S.; Abrahamsen, C. Evaluation of 'the Buddy Study', a peer support program for second victims in healthcare: a survey in two Danish hospital departments. BMC Health Serv Res 2022, 22, 566, doi:10.1186/s12913-022-07973-9.
- Wu, A.W. Medical error: the second victim: The doctor who makes the mistake needs help too. BMJ 2000, 320, 726–727, doi:10.1136/bmj.320.7237.726.
- Mitarbeitersicherheit ist Patientensicherheit: Psychosoziale Unterstützung von Behandelnden im Krankenhaus; Strametz, R.; Aktionsbündnis Patientensicherheit e.V., Eds.; W. Kohlhammer Verlag: Stuttgart, 2021, ISBN 9783170399716.
In Line 74 there is a reference to bank employees suffering to PTSD-like symptoms. However, there is a bundle of phenomena and diseases that all can be summarized under the term "psychosocial burden". There should be some clarification about the different entities that are presented together and addressed by the study program: Second victim, moral injury, moral distress, moral stress, psychological stress, PTSD, burn-out, depression (that has to be diagnosed by a psychiatrist) and many other entities belong to this group and cannot easily be treated as one and may have other coping strategies. Thus, training programs educating peers should give some distinction about this and should be presented more stronger as a "first psychological aid" to health care providers before transfer to a specialist.
Reply to the author: Thank you thank you very much for your comment. We added the following lines in the named section (line94ff): “Of course, different professional groups as well as different entities are not unrestrictedly comparable. Within the topic presented here, there are basically focuses on Second Victim, Moral Injury, Moral Distress, Moral Stress, Psychological Stress, PTSD, Burnout, Depression and many other entities. These cannot be readily treated as a single entity and may have different strategies to deal with. Therefore, training programs that educate peers should make some distinction and can be seem as "first line" help for health care providers before referral to a specialist. Nevertheless, basic approaches to dealing with serious events can be drawn from other contexts for consideration and understanding.”
Line 88: please explain the abbreviations
Reply to the author: Thank you thank you very much for your advice. We added “[…] (keyword PSNV-E= German abbreviation therefore).”
Line 135: in the manuscript there is a short explanation of the curriculum. As the manuscript is about a didactic format the readers do not get much information about the curriculum and its teaching formats itself. Consequently, a reproduction of the study would not be possible. Please consider distinct and structured information (e.g. using the Kerns Analysis or another curriculum developing tool) addressing the "Problem" (this is pointed out), "The Requirements" (What do we need and want?), "Objectives of the Curriculum" (What are the goals?), "Strategies of teaching" (how do we teach?), "Implementations issues" (Who is responsible? Who approves? Who supports? Who has to be informed?) , "Testing and Evaluations" (How and what do we test?) and "Dissimination" (How do we spread our work?). Further, consider to present the curriculum blueprint, teaching and evaluation formats, time tables, needed materials, educators' qualifications and other information, e.g. in the appendices or supplementals to grant the possibility of repetition of the study and to reveal further insight for readers.
Reply to the author: Thank you thank you very much for your remark. We can fully understand this comment. Since this article focuses on different aspects (what content can be taught in a peer training for medical staff, how the training is evaluated by the target group and which people are interested in the training from peer), we have refrained from providing detailed information as suggested. Such a didactic preparation of the contents seemed too weighty to us at this point. Also due to the available scope of the article, we tried to address this aspect under the section "Contents of the teaching modules". To make further information available we added the following line at the end of the section: “[…] For further insight into the content and didactic realization of the training, please contact [email protected].”
MATERIALS
Line 187: is this really a cross-sectional study with evaluation in a pre-post format (line 190)?
Reply to the author: Thank you thank you very much for your question. It was a cross-sectional study with a large questionnaire, so we offered one part before and one part after the training to split the working time. For a better understanding we added the following line “[…] to split the working time.”
Please explain how the recruitment of the participants was done.
Reply to the author: Thank you thank you very much for your remark. We added the following line “[…] The present study emerged from accompanying research on the peer training. The participants of the peer training represent the sample.”
Line 209: You explained that you did a pre-post format (line 190) with use of a bundle of instruments. It is not clear why statistics is accomplished purely descriptive? With a pre-post examination e.g. by GHQ-12 and subjective workload and risks you would be able to do analytical tests on the development of the participants aside from subjective evaluations. If not possible, discuss this issue, as subjective evaluations are not sufficient and questionable to detect whether a training format is successful or not (depends on how do you define success: do people learn and get competent? Or do they think it was good?)
What were your hypotheses? What is tested? How was it tested? Please line this out for better understanding of your work.
Reply to the author: Thank you thank you very much for your remark. The presented study is based on accompanying research. The results are presented to get a first insight of what content can be taught in a peer training for medical staff, how is the training evaluated by the target group and which people are interested in the training from peer. Unfortunately we can’t offer an RCT or any inferential statistic analyses at this point. But looking to the future we are keen to catch up to these points.
RESULTS
As mentioned above the result section is minimalistic as only descriptive data is represented. If there was pre-post data, a significant amount of the results is lost in the present manuscript form. This epidemiological data is described satisfactorily (and is important), although the selection bias is not discussed. Presentation of the data should be improved (why single columns and not boxplots?). With groups and the presented data on mean and SD effect sizes (Cohens D or even Cohens D-av for inter-, Cohens D-z for intra-individual calculations) are calculatable and relevant for educators, curriculum planers, quality managers and distributors.
Reply to the author: Thank you thank you very much for your remark and suggestions. Due to the aspect that we did only accompanying research and focused for a first insight on different aspects (what content can be taught in a peer training for medical staff, how the training is evaluated by the target group and which people are interested in the training from peer) we are keen to catch up to the raised points.
To address the selection bias, we added the following line in the section “Discussion”: “[…] Limitations to note are that the sample studied is not yet of a truly robust size and might show a selection bias. […]”.
Line 349
Ratings of the programs to be "good" or "Not good" (x-axis labeling is not explained) are highly (!) subjective and not robust for format or curriculum evaluation and therefore are not an indicator of effectiveness, efficacy, or efficiency of the program. Please discuss.
Further, there is no information whether learning objectives have been accomplished or not – if there were any outlined: what were the measurable (!) objectives and goals of the program? Were there tests on learning dimensions and competence such as factual knowledge, attitude, communication skills (see Bloom's taxonomy)? If not: discuss it for future and ongoing research as they are quality indicators of curriculum evaluation.
What Bias do you expect? To my eyes: selection bias (e.g. motivation to be trained, perhaps fees and social desirability.
Further, subjective stress from 2017 to the pandemic may have changed. Also, national regulations changed (eg. PPUGV in Germany on the one side, staff shortages of the other). Please discuss these variations. Has psychological burden changed in participants in 2017 to 2020 ?
Have there been subgroups effects, e.g. in the stress-groups or workload-groups? Age depend changes?
Reply to the author: Thank you thank you very much for your remark and suggestions.
We have supplemented the labelling of the response scale in both the methods and results sections.
We added a whole section “Limitations” to address different aspects:
Limitations.
Limitations to note are that the sample studied is not yet of a truly robust size and might show a selection bias (e.g. motivation to be trained, tuition fees for attendance, social desirability). In addition, only initial descriptive analyses have been conducted so far. In this first analyses we offered subjective ratings to the participants which can only be used to answer the question if the participants “feel” more informed, educated, and skilled instead of testing their competencies. Future research is planned to focus e.g. on measurable objectives as learning outcome. For selected aspects, more in-depth analyses should provide an insight into the profiles of people who want to be trained as peers in order to strengthen the current findings that interested persons have a suitable personality profile / stress profile / coping profile ("which group of people decides to be trained" combined with the low-er-than-average GHQ scores of participants may suggest that those who chose to participate had the psychological resources to take part").
Overarching developments with an influence on the health care system can be mentioned as limitations. In the period between 2017 and 2020, there may have been different influencing factors that affected the willingness to participate in peer education, the experience of stress, increasing staff shortages, or other factors.
For the question of which people a clinic/company should select to be trained as a peer, there are currently only empirical values (please contact www.psu-akut.de). Further studies are planned.
DISCUSSION
Throughout the discussion more external validity would be helpful for better stratification of generalization of some conclusions. However, as analytical data seems to be available, the discussion section should revisited when analytics were done. If not, this should be discussed in more detail.
Further, as the main hypotheses are not outlined, readers may be confused by this discussion as it is mainly epidemiological and focusing on descriptions but not on the learning format and the program that is the protagonist of the manuscript at the first view. Consequently, the heading should be adopted in case epidemiological findings alone are the aim of this study. Otherwise it may be distracting for some readers.
Reply to the author: Thank you thank you very much for your remark and suggestions. We tried to address this point in several sections (abstract, introduction, discussion, limitations, conclusion) to make the main focus of the article more visible.
Line 406: Please discuss this gender bias.
Linie 414: This may be prone to a selection bias of the sample.
Reply to the author: Thank you thank you very much for your remark. We addressed both aspects in the Limitations.
Why do you think that a low test results in a single GQH-12 is a valid and reliable parameter for resilience or psychological robustness? Are there studies on "robust" persons with the score? I think we should be careful, as psychiatrist that may be considered very robust can be second victims too and psychological stress may even effect robust people depending on the situation.
Reply to the author: Thank you thank you very much for your comment. In chapter 2.2.1 we tried to describe why the GHQ can be seen as a reliable and valid instrument. In the chapter “Discussion – Regarding question 3 – Do the peers bring the desirable psychological robustness” we compared the GHQ results of the sample in our study with GHQ results with a sample of general population and further results. We agree with you that even people with robust GHQ values can become second victims.
CONFLICT OF INTEREST
Please reevaluate any COI or bias arising from it as the format reported on is a "semi-commercial" product according to the web-page and the study group is the only provider of it.
Reply to the author: Thank you thank you very much for your remark. We added a paragraph about this in the chapter conflicts of interest:
Financial disclosure statement: The results presented here were obtained from accompanying research on peer training conducted by PSU-Akut e.V.. Further information on the financing of the training is presented in the following.
The training costs 320€ per participant per day during the survey period. (For the year 2023, the tuition fee was increased to 350€ per participant per day due to increased energy costs and thus higher rents, catering costs, etc.). This money was used to finance all setting expenses (rent of rooms in a metropolitan area, catering, teaching-learning materials, certificates, pro-rata staff position for coordination, contact and registration management, advertising and public relations, etc.) as well as honoraria for the lecturing trainers. The lecturing trainers are experts with many years of experience in psychosocial support, psychotraumatology and crisis intervention, who have many years of practical experience (e.g. crisis intervention) as well as experience in the clinical field (e.g. doctors). Due to the small group sizes required for the training, no profit was made on the participant fees. All tuition fee is only used for reimbursement of training including the paid honoraria.
The conceptual, technical and methodological-didactic development of the peer training was funded by the Bavarian Medical Association (legal professional representation of all Bavarian physicians in Germany, Federal State of Bavaria) and the Bavarian State Ministry of Health and Care (StMGP).
The accompanying research was supported by third-party funding from the Professional Association for Health Services and Welfare Care (BGW).
Of the authors, the following individuals served as trainers in the peer education and received honoraria: Andreas Schießl, Andreas Igl
For the participating medical staff, a sponsorship was established by the Municipal Accident Insurance Bavaria (Institution for Statutory Accident Insurance in Bavaria, Germany). The participants pay (if defined criteria are met) only 50% of the course fees. The other 50% is covered by these subsidies up to a defined amount. The non-profit organizer PSU-Akut e.V. has no monetary advantage and coordinates this funding for the participants.
In each course there were individual participants, who could take part without a financial contribution if certain criteria were met. Thus, the orientation of a non-profit organization was taken into account.
As a conclusion: I think Hinzmann et al conducted a pioneering and promising approach to create a peer support program. Unfortunately, to my opinion this manuscript does not use its full potential.
The authors should consider presenting concise information about the curriculum, outline hypotheses, should do deeper analytics on available data, reevaluate graphics and calculations of effect sizes and should reevaluate the whole discussion and conclusion section after that.
Reply to the author: Thank you for your feedback that you see so much potential in our study. We have endeavored to revise all the points we could.
We hope you understand our adjustments, thank you again for your efforts and support, and remain available to answer any queries you may have!
Kind regards,
Dominik Hinzmann, [email protected]

Round 2
Reviewer 2 Report
Thank you for all amendmends.
Still, I am not completely convinced why comparing data cannot be published and thus I would like to encourage the authors to do so OR to lable the study as a "Pilot Study" in the titel.
Otherwise I do not have further comments.
Author Response
Dear Reviewer,
Thank you very much for your renewed feedback and constructive suggestions.
“Still, I am not completely convinced why comparing data cannot be published and thus I would like to encourage the authors to do so OR to lable the study as a "Pilot Study" in the title.”
We are very sorry that we could not reach you regarding the comparative data.
In the spirit of scientific discourse, we propose to change the title of the publication as follows.
First results of Peer Training for Medical Staff - Psychosocial Support through Peer Support in Health Care
We hope that we have done sufficient justice to your comments and would like to thank you once again for your critical and important comments. We are happy to be at your disposal for further exchange beyond this review process.
King regards,
Doominik Hinzmann, [email protected]
